# The Effect of the Reaction pH on Properties of Lead(II) Azide

**DOI:** 10.3390/ma14112818

**Published:** 2021-05-25

**Authors:** Jolanta Biegańska

**Affiliations:** Department of Hydrogen Energy, Faculty of Energy and Fuels, AGH University of Science and Technology, 30-059 Cracow, Poland; biega@agh.edu.pl; Tel.: +48-12-617-20-26

**Keywords:** lead(II) azide, nitrate-based method, acetate-based method, pH of chemical reactions, size of crystals

## Abstract

Lead(II) azide is an initiating explosive; even a small amount can trigger an explosion caused by simple external stimuli, such as sparks, flames, friction or pinpricks, and is able to initiate the explosive reaction of rock-crushing explosives. Due to the fact that this initiating explosive triggers further reactions, the effect of priming detonators depends on the properties of its material. Its sensitivity is associated with the size of its crystals. For instance, it is used for mining detonators in the form of fine crystals. The quality of the crystals is also correlated to the safety of the production process, i.e., the crystals should be round-shaped rather than needle-like since breaking it would inevitably trigger an explosion. The process of lead(II) azide production on an industrial scale is based on the reaction of lead(II) nitrate with sodium azide with the presence of dextrin, which determines the desired shape of the crystals. The reaction pH affects the number of sediment particles formed in a periodical reactor. Changing the pH from 6.5 to 7.5 leads to the rapid growth of crystal particles.

## 1. Introduction

According to the SIC BREF (Production of Specialty Inorganic Chemicals—SIC) [1], the production of explosives is based on three lead compounds that have key industrial and economic importance for Europe, namely, lead azide (PbN_6_), lead trinitro resorcinol (C_6_HN_3_O_8_Pb) and lead picrate (C_6_H(NO_2_)_3_O_2_Pb·H_2_O). Nevertheless, lead(II) azide is the compound that is most frequently used to initiate detonations of crushing explosives and is manufactured in ten EU member states: the Czech Republic, France, Spain, Italy, Germany, Portugal, Austria, Sweden, the U.K. and Poland [2]. It exhibits a high degree of sensitivity to impact, friction and electrostatic discharge, and even a small amount can trigger the detonation of crushing explosives [3,4]. The sensitivity of this compound is related to the size of its crystals [5,6,7,8,9,10]. The quality of the crystals is also very important for the safety of manufacturing since uniform and round-shaped crystals (with no caverns or dust additives) enable easy pouring (filling of priming detonators) to achieve a sufficient granulometric weight.

The process of lead(II) azide manufacturing on an industrial scale is based on the reaction between lead(II) nitrate and sodium azide (Figure 1). Solutions of these salts at appropriate concentrations, together with necessary admixtures (dextrin, crystallization seeds), are, to some degree, determinant factors for achieving the desired properties of crystals.

The purpose of the research described in this article was to develop a method of producing lead(II) azide. This required determining the appropriate process parameters (pH of the reaction environment, reagent dosing method) that would enable obtainment of the product with specific performance properties (low sensitivity to friction and appropriate form of the product—i.e., the crystals should be round-shaped rather than needle-like). The specificity of this explosive material—a highly sensitive material—required an unconventional approach to find the solution of the problem. Tests were conducted on a laboratory scale, maintaining requirements reflecting the process. The mass of the product manufactured (up to 5 g) and the test method adapted to this type of explosive (characterized by a minimum of contact and ensuring safe operation) were limited.

The following types of lead azide [16,17,18,19,20,21,22] are distinguished (see Table 1).

Lead(II) azide is considered to be a highly sensitive explosive and has been classified into the first group of safety precautions [23]. Some explosive properties of lead(II) azide are summarized in Table 2.

Crystals of lead(II) azide are white or slightly yellowish in the form of irregular needles—see Figure 2:

Lead(II) azide is a polymorphic compound since it can take four forms, which are alpha, beta, gamma and delta [24]. Experimental values of the sensitivities of α, β, and γ-forms were published by Wyatt, Wythes, Taylor and Thomas [4,25,26]. The gamma modification is more sensitive to impact but less sensitive to friction than the α and β forms. The electrostatic discharge sensitivity is about the same for all three polymorphs.

The colloidal form, called alpha, is obtained [27] when the manufacturing process is carried out with intense stirring and the admixture of highly concentrated reagents. However, some efforts are reported [27,28,29] on the production of its phlegmatized variety.

The beta form of lead(II) azide is produced, using a method that consists of a slow diffusion of lead ions with sodium azide. It enables the production [30,31,32] of long and extremely sensitive needles prone to spontaneous explosion. Due to safety issues with the product application, the aforementioned manufacturing method was abandoned as unsuitable for practical use.

The gamma and delta forms of lead(II) azide can be produced when some restrictions for the pH of solutions are maintained [3,24,33], i.e., pH = 3.5–7.0 for the gamma form and pH = 3.5–5.5 for the delta option.

## 2. Materials and Methods

### 2.1. Thermodynamic Analysis of Chemical Reaction

Lead(II) azide can be produced when the following reactions take place:(1)Nitrate method;
(1)$$2NaN_{3}+Pb{\left(NO_{3}\right)}_{2}=Pb{\left(N_{3}\right)}_{2}+2NaNO_{3}$$
and(2)Acetate method;
(2)$$2{\mathrm{NaN}}_{3}+\mathrm{Pb}{\left({\mathrm{CH}}_{3}\mathrm{COO}\right)}_{2}=\mathrm{Pb}{\left(N_{3}\right)}_{2}+2{\mathrm{CH}}_{3}\mathrm{COONa}$$

The analysis of the thermodynamic conditions of both methods is summarized in Table 3.

The values of the parameters listed in Table 3 demonstrate that both methods (nitrate and acetate one) lead to the release of similar amounts of free enthalpy (∆H°) and thermodynamic potential (∆G°). In addition, the equilibrium degree of the conversions (α) is very close to 1 in the case of both methods, regardless of the method of lead(II) azide production. It means that the reaction is irreversible and lasts until all substrates are consumed.

A strong similarity between the parameter values obtained for both methods proves that the thermodynamic conditions for the production of lead(II) azide are practically identical for both methods.

### 2.2. Analysis of Reaction Ambient Conditions

The process of the lead(II) azide production on an industrial scale must be conducted under specific conditions concerning the pH of the initial solutions:
(1)Solution of Pb(NO_3_)_2_, pH = 1.89–2.15;(2)Solution of NaN_3_, pH~10–11.

The theoretical evaluation of the reaction pH was carried out under the assumption that the process of lead(II) azide precipitation is conducted under stoichiometric conditions and that substrates react with one another in the following proportions (Table 4).

Table 5 includes a summary of the results obtained for selected parameters of both methods.

The results listed in Table 5 indicate that the result of the reaction pH calculated for the acetate method is, on average, higher by 1.6 units than that for the nitrate method when the same ratios of reagents are assumed (reactions No. 4 to 8). Slight differences, roughly by 0.2 units, are noted for fractions with a large surplus of NaN_3_ with respect to Pb(NO_3_)_2_ or PbAc_2_ (reactions 1–3). The increase in the c_1_/c_2_ ratio between the concentrations leads to a decrease in pH (Figure 3), where a substantial change, i.e., rapid drop of pH, is recorded in the case of both methods when the respective A:B ratios between reagents are 1:3 and 1:2 (when concentration ratios c_1_/c_2_ range from 0.33 to 0.5).

### 2.3. Production of Lead(II) Azide

Distilled water was poured into a reactor tank made of polypropylene, and strictly measured amounts of reagent solutions, i.e., 0.1 mol solution of lead nitrate (or acetate) and 0.1 mol solution of sodium azide were added with the blade stirrer continuously rotating (Figure 4).

Amounts of solutions added to the reactor depended on the reaction course—ratios between reagent volumes (reactions No. 1, 2, 3, 4, 5, 6, 7, and 8) were specified so as to produce not more than 5 g of lead(II) azide during each experiment.

The method of reagent injection was properly adjusted to enable simultaneous consumption of substrates. After the whole (measured) amounts of reagent solutions were added and stirred, the stirrer was switched off and a drop of the resultant solution with lead(II) azide suspended therein was taken from the tank for further evaluation of the particle number in a Thoma chamber.

The lead(II) azide sediment was extracted from the solution with a Büchner funnel whilst the solution eluate was taken for the determination of pH for the post-reaction solution. Then, the sediment of lead(II) azide was flushed with distilled water and methyl alcohol. After bleeding filtration, the lead(II) azide solution was evaporated at the temperature of 333 K for 1 h. Then, the sediment was cooled down and samples of the material were taken for determination of its sensitivity to friction and bulk density.

Remnants of the lead(II) azide were decomposed in nitric acid with an admixture of sodium nitrate.

#### 2.3.1. pH Test of the Filtered Eluate

After having the pH meter calibrated, its electrode was washed several times with double distilled water and then with the solution during the test. After the electrode was perfectly washed, it was immersed in a glass beaker filled with the investigated solution and then, after several minutes of delay, the result was read from the instrument.

#### 2.3.2. Evaluation of the Particle Number in the Sediment

A drop of the post-reaction solution was taken from the bottom of the vessel with a glass rod and placed into a clean Thoma chamber. Then, the chamber was placed under a microscope and the number of particles was counted in 40 small squares with the microscope magnitude of 10 × 10. Finally, the number of particles in 1 cm^3^ of the solution was calculated on the basis of the following formula:(3)$$\mathrm{umber}\;\mathrm{of}\;\mathrm{particles}\;\mathrm{in}\;1\;{\mathrm{cm}}^{3}=\frac{A}{40}\times 4000=A\times 10^{5}$$
where A—Number of particles in 40 small squares.

#### 2.3.3. Investigation of Sensitivity to Friction

The sensitivity to friction was investigated in line with the applicable standard [38]. Such investigations were carried out in a Julius Peters apparatus. The friction sensitivity was expressed as the lowest impact pressure of a rod depressed to a plate, triggering at least one explosion or deflagration of the material in six consecutive tests (or the maximum frictional force at which no reaction was recorded in six consecutive tests).

#### 2.3.4. Determination of Bulk Density

The bulk density was determined in line with the applicable standard [39]. A cylinder with the capacity of 5 cm^3^ was weighed with an accuracy of 0.01 g and then dried lead(II) azide was poured inside through a funnel made of smooth paper and protected with a special screen. The cylinder was filled up to the level marked with a line and weighed once again. The bulk density ‘x’ was calculated according to the following formula:(4)$$x=\frac{m_{2}-m_{1}}{5},\;g\cdot {\mathrm{cm}}^{-3}$$
where m_1_—Weight of the empty cylinder (in g), m_2_—Weight of the cylinder filled with lead (II) azide (in g), 5—Cylinder capacity (in cm^3^).

#### 2.3.5. Size and Shape of Lead(II) Azide Particles

Some pictures of collected sediments were taken to illustrate the shapes of lead(II) azide particles [40,41].

## 3. Results and Discussion

### 3.1. Measurement of pH of the Filtered Eluate and Solutions

The results of pH measurements carried out for post-reaction solutions are summarized in Table 6 and compared against the theoretical expectations provided in the adjacent columns.

The numerical results of pH parameters obtained from the experiments reveal only slight differences as compared to the theoretical expectations with a mean deviation as low as about 0.3 units. It serves as a practical confirmation of the theoretical calculations.

The increase in the c_1_/c_2_ ratio leads to a decrease in the pH parameter for the reactions (Figure 5), although a rapid drop of pH is recorded for the nitrate method when the c_1_/c_2_ ratio increases from 0.3 to 1.0 whilst for the acetate method, the similar drop takes place for the c_1_/c_2_ increase from 0.3 to 0.5. A twofold drop in pH occurs in the case of the nitrate method.

### 3.2. Evaluation of the Particle Number in the Sediment

The results for the reaction productivity and the values for the parameter that represent the granulation of lead(II) azide particles (number of particles in 1 cm^3^∙10^6^) are summarized in Table 7.

The efficiency is deemed satisfying in both cases, although the efficiency levels obtained for the nitrate method are better by several percentages. The impact of the reaction pH is insignificant in that case (see results in Table 6), although the pH level substantially affects the granulation of sediments originating from the reaction, expressed by the number of lead(II) azide in 1 cm^3^ of the measurement chamber, which is illustrated in Figure 6.

The number of lead(II) azide particles formed during reactions No. 4 to 8 is quite similar; however, an increase in the reaction pH from about 5.3 to 6.4 leads to a remarkable decrease in that parameter with an insignificant increase in the size of the particles. The number of lead(II) azide particles significantly increases for reaction No. 3, which is observed in the case of both methods, with the simultaneous formation of a huge number of tiny forms. It corresponds to an increase in the reaction pH, ranging from 6.5 to 7.4 when the number of particles recorded in the sediment increases by about ten times. The curve plotted for the illustration of the relationship between the number of particles and the reaction pH for the nitrate method within the pH range from 6.5 to 7.0 overlaps the respective curve plotted for the acetate method.

The value of the pH of the reaction has a significant effect on the granulation of the resulting product, characterized by the number of particles of lead(II) azide in 1 cm^3^ of the measuring chamber. As the pH value changes, with a small excess of sodium azide in relation to the stoichiometric ratios, there is a step change in the quantity of lead(II) azide “crystals” formed. The acetate method produces long forms that can detonate automatically under the effects of crystal friction.

### 3.3. Investigation of Sensitivity to Friction

The sensitivity to friction of lead(II) azide was the same, regardless of the production method since the load of 10 g was obtained in the case of both (nitrate and acetate) methods.

### 3.4. Determination of Bulk Density

The bulk density of lead(II) azide produced by means of the nitrate method was 0.8 g/cm^3^ with 0.7 g/cm^3^ for the same compound produced using the acetate method.

### 3.5. Size and Shape of the Lead(II) Azide Particles

Some pictures (Figure 7a–d) of selected sediments collected from the reaction products were taken to illustrate the shapes of lead(II) azide particles [40,41].

The sizes of the lead(II) azide particles varied. The largest in size, but the smallest in number, in 1 cm^3^ were the particles obtained in reaction No. 4 with the use of the nitrate method (Figure 7a) and acetate method (Figure 7c). Lead(II) azide particles produced by means of the acetate method had sharp edges, which is particularly evident in Figure 7c,d. It is the reason why these particles were extremely sensitive to friction.

## 4. Conclusions

(1)Regardless of the method of lead(II) azide production (nitrate or acetate method), the chemical reaction is irreversible until the reaction substrates are completely consumed. The foregoing is proved by the value of the equilibrium degree of conversions (α), which is very close to 1 in the case of both methods.(2)For both the nitrate and acetate method, the reaction pH strongly depends on a mutual ratio between the reagents in the range below 0.3 and 0.5, i.e., the only range that is suitable for industrial applications due to safety reasons.(3)Some effect of the reaction pH on the number of sediment particles formed in a periodical reactor was recorded. For both the nitrate and the acetate method, any change of the reaction pH within the range from 6.5 to 7.5 leads to a rapid increase in crystal particles originating from the reaction, which confirms that the flow of both reagents or the pH of their solutions must be stabilized.

## Figures and Tables

**Figure 1 materials-14-02818-f001:**
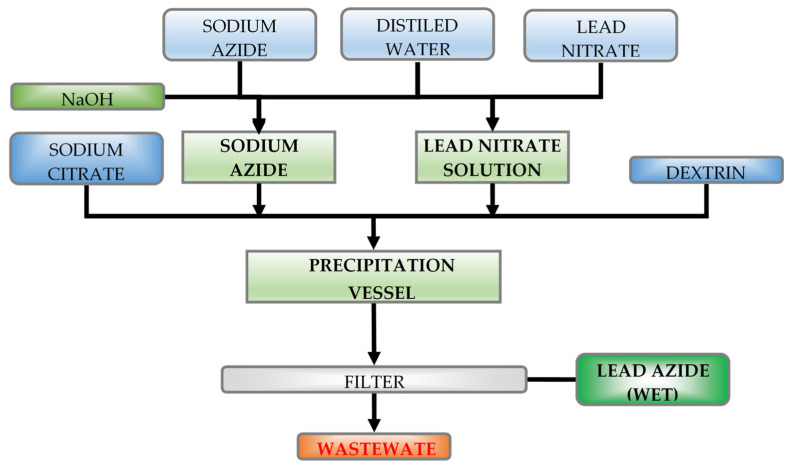
Manufacturing process of lead azide (based on [11,12,13,14,15]).

**Figure 2 materials-14-02818-f002:**
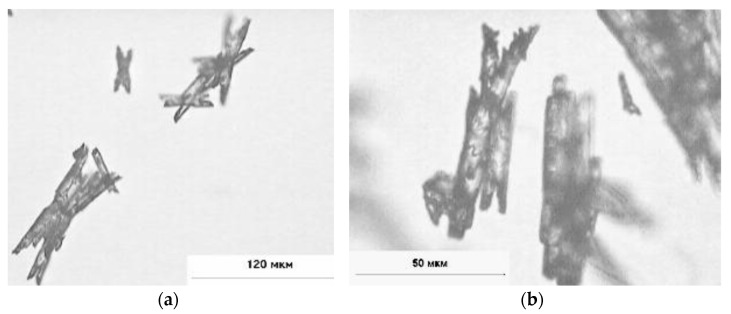
Lead(II) azide crystals in the form of needles, (**a**) magnification of 400×, (**b**) magnification of 1000×. Reprinted from ref. [17].

**Figure 3 materials-14-02818-f003:**
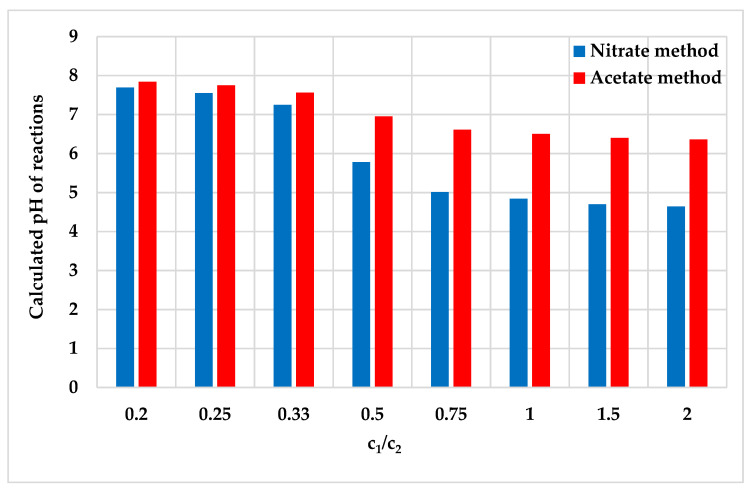
Relationship between theoretical (calculated) pH parameter of the reaction and c_1_/c_2_ ratios between analytic concentrations of reagents.

**Figure 4 materials-14-02818-f004:**
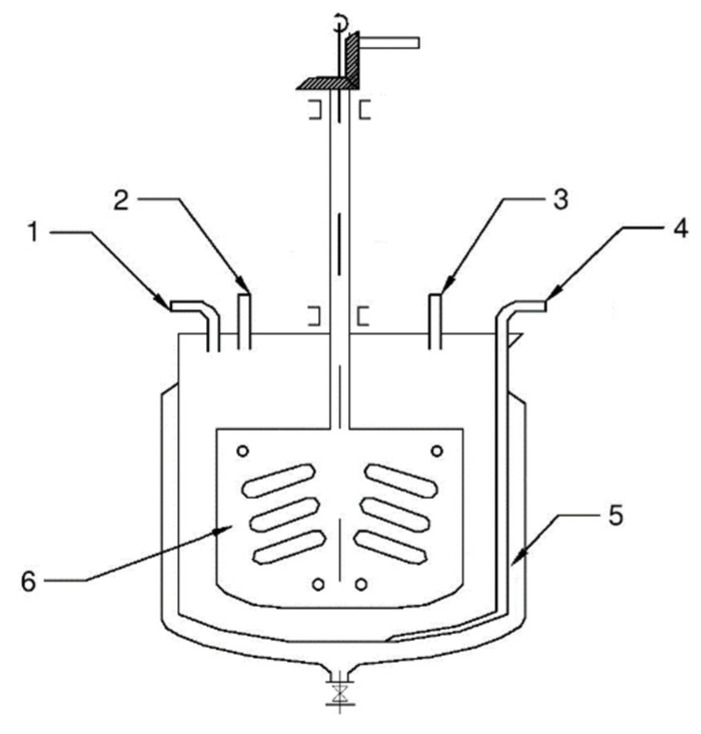
Schematic diagram of a reactor for production of lead(II) azide using the periodical method. Description: 1–3—Inlet tube, 4—Siphoned pipe for discharge of solutions from the reactor, 5—Water jacket for cooling or heating of the reactor content, 6—Stirrer.

**Figure 5 materials-14-02818-f005:**
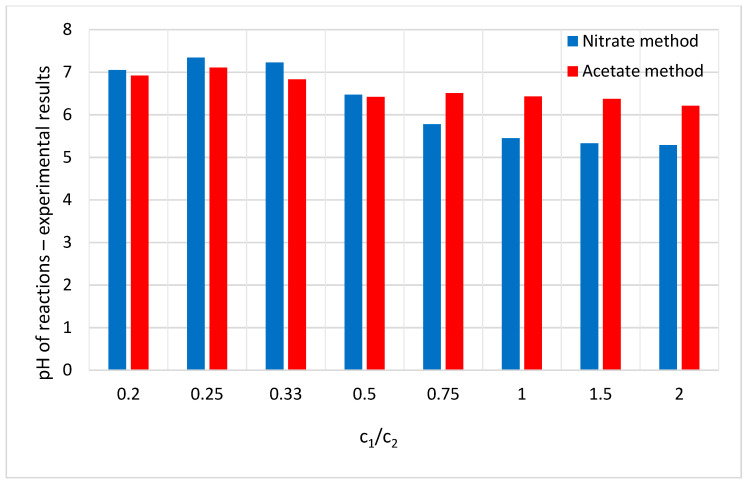
Relationship between experimental results for the pH parameter of the reactions and c_1_/c_2_ ratios between analytic concentrations of reagents.

**Figure 6 materials-14-02818-f006:**
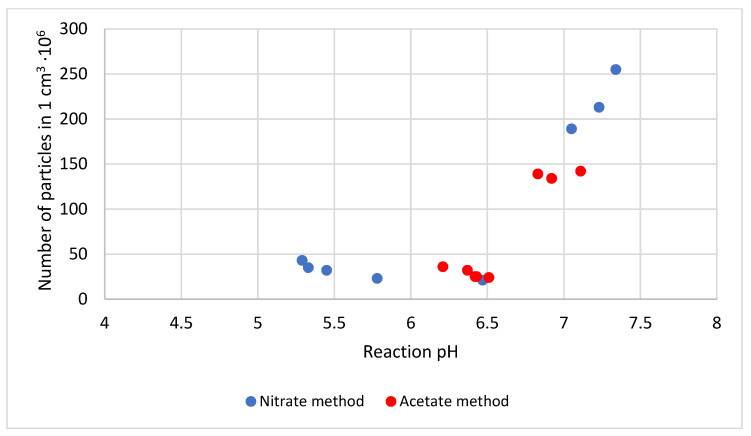
Relationship between the number of particles and the pH of the lead(II) azide forming reaction.

**Figure 7 materials-14-02818-f007:**
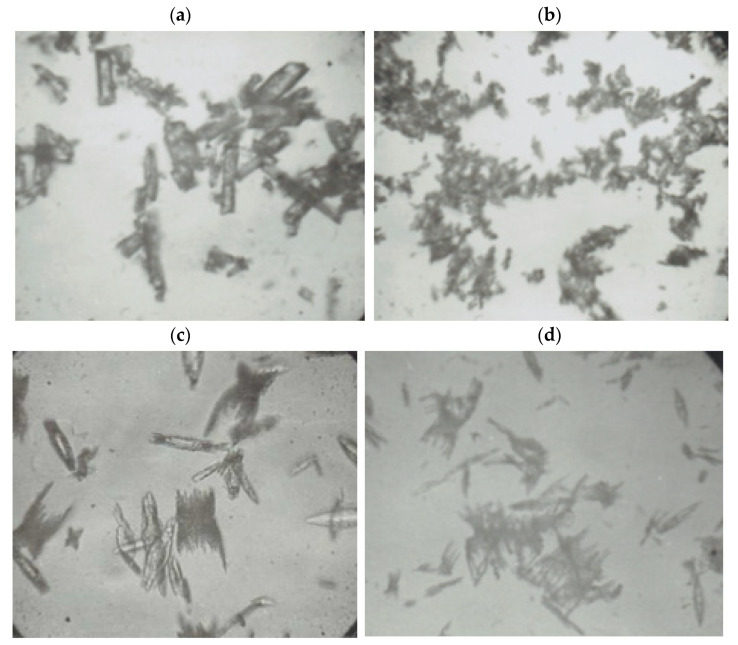
(**a**–**d**) Pictures of lead(II) azide particles (crystals).

**Table 1 materials-14-02818-t001:** Main types of lead(II) azide suitable for industrial applications. Reprinted from ref. [19,22].

Type of Lead Azide	Product Characteristics
Service (SLA) [16,17,18]	Service lead azide (SLA) was developed by the British and does not involve a coating agent; instead, it makes use of the addition of acetic acid and sodium carbonate, which provides a nucleus to precipitate lead azide in a less hazardous (compared to needle-shaped crystals) spherical morphology. The SLA has higher explosive performance than dextrinated lead azide (DLA) or RD1333/special purpose lead azide (SPLA) but is somewhat more sensitive.
Colloidal [19]	Colloidal lead azide is the purest form of lead azide used in the United States and is made by mixing dilute solutions of lead and azide salts. It has a very fine particle size of ~5 μm, making it ideal for its main application, which is coating on electric bridgewires for commercial electric detonators. It is known to be extremely sensitive to electrostatic discharge (ESD).
Dextrinated (DLA) [16,17,18]	Dextrinated lead azide (DLA) is considered to be the safest-to-handle form and the most common type now used commercially. It was developed in the United States in 1931 as a solution to the numerous accidental explosions associated with attempts to manufacture pure lead azide. Its key feature is the incorporation of dextrin (a short-chained, starch-based polysaccharide), which helps to desensitize the explosive by preventing the formation of large fragile crystals, albeit at the cost of performance and added hygroscopicity.
RD1333 (SPLA) [16,17,20]	RD1333 and special purpose lead azide (SPLA): Although involving somewhat different processing pathways, both British-developed RD1333 and the later U.S.-developed SPLA are similar in that they have nearly identical performance requirements/specifications and make use of the sodium salt of carboxymethylcellulose as the desensitizing agent. As such, in the United States, the two types are often used interchangeably. The materials were developed to meet the need for lead azide, which performed better than DLA (especially in smaller detonators), while maintaining some of its safe handling characteristics. The SPLA makes up a large portion of the current U.S. military stockpile of lead azide, which has existed since the 1960s.
On-demand (ODLA) [21]	On-demand lead azide (ODLA) is a much more recent development compared to the others described in this report. It is a military-qualified process developed by the U.S. Army Armament Research, Development and Engineering Center, Picatinny Arsenal, NJ, which produces lead azide that meets the RD1333 specification and is considered to be the equivalent to RD1333 and SPLA. The main advantage is that it is produced in an on-demand, continuous fashion, thereby avoiding the hazards associated with handling large batches of the material. The small footprint and low cost of the processing equipment also means that it can be placed close to item production lines, further reducing the need for expensive and dangerous transport of lead azide on public roads. The ODLA was qualified by the U.S. Army in 2012 and is currently being evaluated in larger-scale loading operations.

**Table 2 materials-14-02818-t002:** Basic explosive properties of lead azide compared to lead styphnate. Reprinted with permission from ref. [22].

Material	Impact (J)	Friction (N)	ESD (mJ)	Density (g/cm^3^)	DSC (°C)	VOD (m/s)
**Lead azide**	0.089	<1	5.0	4.80	315	5500
**Lead styphnate**	0.025	<1	0.2	3.00	282	5200

Abbreviations: ESD—Electrostatic discharge; DSC—Differential scanning calorimetry; VOD—Velocity of detonation.

**Table 3 materials-14-02818-t003:** Numerical results of thermodynamic calculations. Adapted from ref. [34,35,36,37].

Parameter	Symbol and Unit	Method	
		Nitrate	Acetate
Gibbs free energy (total heat)	(∆H°) (kJ/mol)	−1202.9	−1175.3
Thermodynamic potential	∆G°(kJ/mol)	−1212.9	−1462.6
Constant of chemical balance	K_a_	542.5 × 10^210^	34.168 × 10^255^
Equilibrium degree of conversions	α	0.9999	0.9999

**Table 4 materials-14-02818-t004:** Contribution of substrates to the reaction pH calculation.

Parameter and Ratio		Reaction Number							
		1	2	3	4	5	6	7	8
A	a, o	1	1	1	1	1.5	2	3	4
B		5	4	3	2	2	2	2	2

A—Pb(N_3_)_2_ or Pb(CH_3_COO)_2_ ≡ Pb(Ac)_2_, B—NaN_3_.

**Table 5 materials-14-02818-t005:** Theoretical results of the reaction pH for the nitrate (a) and acetate (o) method.

Parameter and Methods		Reaction Number							
		1	2	3	4	5	6	7	8
V	a, o	6	5	4	3	3,5	4	5	6
c_1_/c_2_	a, o	0.2	0.25	0.33	0.5	0.75	1	1.5	2
p_a_H	a	7.69	7.55	7.25	5.78	5.01	4.84	4.70	4.64
	o	7.84	7.75	7.56	6.95	6.61	6.50	6.40	6.36
{H^+^}	a	2.03 × 10^−8^	2.81 × 10^−8^	5.61 × 10^−8^	1.65 × 10^−6^	9.8 × 10^−6^	1.45 × 10^−6^	1.98 × 10^−5^	2.30 × 10^−5^
	o	1.43 × 10^−8^	1.77 × 10^−8^	2.76 × 10^−8^	1.11 × 10^−7^	2.49 × 10^−7^	3.19 × 10^−7^	3.95 × 10^−7^	4.37 × 10^−7^
n	a	9.98 × 10^−2^	9.98 × 10^−2^	9.78 × 10^−2^	9.59 × 10^−2^	9.82 × 10^−2^	9.816 × 10^−2^	9.788 × 10^−2^	9.754 × 10^−2^
	o	9.96 × 10^−2^	9.95 × 10^−2^	9.92 × 10^−2^	9.07 × 10^−2^	9.31 × 10^−2^	9.56 × 10^−2^	9.56 × 10^−2^	9.52 × 10^−2^

V—Volume of the reaction solution after having the reagents mixed together; c_1_—Analytic concentration of $\mathrm{Pb}{\left({\mathrm{NO}}_{3}\right)}_{2}$ or PbAc_2_, after having the reagents mixed together; c_2_—Analytic concentration of ${\mathrm{NaN}}_{3},$ after having the reagents mixed together; $p_{a}H$—Hydrogen ion exponent for calculated for $\left\{H^{+}\right\}$, $p_{a}H=-\log\{H^{+}\};$ $\left\{H^{+}\right\}$—Activity factors; n—Number of $\mathrm{Pb}{\left(N_{3}\right)}_{2}$ moles precipitated in the form of sediments.

**Table 6 materials-14-02818-t006:** Results of pH measurements for the reaction of lead(II) azide formation.

Reaction Number	Numerical Results for the Methods Involved			
	Nitrate		Acetate	
	Calculated	Experimental	Calculated	Experimental
1	7.69	7.05	7.84	6.92
2	7.55	7.34	7.75	7.11
3	7.25	7.23	7.56	6.83
4	5.78	6.47	6.95	6.42
5	5.01	5.78	6.61	6.51
6	4.84	5.45	6.50	6.43
7	4.70	5.33	6.40	6.37
8	4.64	5.29	6.36	6.21

**Table 7 materials-14-02818-t007:** Results of measurements of the reaction efficiency and number of lead(II) azide particles.

Reaction Number	Method			
	Nitrate		Acetate	
	Reaction Efficiency, %	Number of Particles in 1 cm^3^∙10^6^	Reaction Efficiency, %	Number of Particles in 1 cm^3^∙10^6^
1	95.50	189	96.47	134
2	91.74	255	90.07	142
3	94.31	213	88.03	139
4	90.18	21	85.88	25
5	94.16	23	90.05	24
6	93.02	32	84.22	25
7	92.60	35	82.54	32
8	93.34	43	82.94	36

## Data Availability

Data available in a publicly accessible repository. The data presented in this study are openly available.

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
