# Peer review of "The Effect of the Reaction pH on Properties of Lead(II) Azide"

_materials, 2021, doi:10.3390/ma14112818_

Round 1

Reviewer 1 Report

The author reports a systematic characterisation of the morphology of Lead(II) azide crystals  obtained via two synthetic reactions (methods) as a function of pH and reagents concertation.

% Introduction

In the introduction the author should specify with respect to what lead(II) azide crystals exhibit a high degree of sensitivity (line 27).  The sensitivity of lead(II) azide is indeed a recurring concept in the manuscript and thus its meaning should be clarified. Is the author referring to sensitivity to friction? Please state explicitly.

References should be added to support the author’ statement that the sensitivity of lead(II) azide depend on size and quality of the crystals (line 28-29).

Have all polymorphs of lead(II) azide (alpha, beta, gamma and delta) relevant explosive properties? Specify this in the introduction when discussing these polymorphs (line 53-55). In this respect specify also to which polymorph corresponds the crystal description reported at line 47-49 and shown in Figure 2.

Add to the introduction the scope of this research and highlight more clearly the current limits in the field that this research aims at addressing.

% Materials and Methods

Provide citations for the calculation of the thermodynamic parameters reported in table 3.

Report formula used for the calculation of the pH listed in table 4 and clarify meaning of the labels: V, c1/c2 etc. Add units where relevant.

% Results and Discussion

In figure 6 the definition of the “fitting” function is not reported and therefore the lines connecting the points have no meaning and should be removed. Even as a guide for the eye these lines are neither acceptable nor satisfactory.

The author describes clearly the results but does not offer an interpretation. How does the pH affect the kinetic of crystallisation seeding? Why the sharpness of particles affects their sensitivity to friction?

Minor Comments:

Line 24-25 “…that is the most frequently used …”

Line 34 “nitrate(VI)” it is not a correct nomenclature. Remove the ‘(VI)’ or explain in the text. I note that the name ‘nitrate’ already specifies the oxidation state of nitrogen in this anion, which is 5+.

Report citation for the industrial synthesis of lead(II) azide at line 34: [3-6]

Line 93-95 are not very clear and readable, consider making a table instead.

Line 133 “..solution was dried ..” consider changing ‘dried’ for ‘evaporated’ or is it meant that the filtrated lead(II) azide product was dried? Please, rephase to clarify the meaning.

Line 147, 160 “.. cm3 ..” the 3 should be a superscript: cm3

Author Response

The answers to all comments point by point have been included in the attached docx file.

Reviewer 2 Report

The manuscript of Dr. Biegańska describes the results of the investigations of experimental conditions for the safety production of lead(II) azide which may be applicable for a manufacturing process. The manuscript is well-written and easy-to-read. I may advice author to include/add more journal references in Introduction which will be easier accessible to readers then the U.S.A. patents.

Some small moments:

  • lines 15 and 34. "...lead(II) nitrate(VI) with sodium azide..."   symbol "IV" is extra
  • line 48. "...lead(II) nitrate(V)..."   (V) is extra
  • line 137.    (III) is extra
  • lines 63-65. "The gamma and delta forms of lead(II) azide can be produced when some restrictions for pH of solutions are maintained [15, 22], i.e. pH = 3.5–7.0 for the gamma form and pH = 3.5–5.5 for the delta option." Check please, seems to be a big range of pH (?)

Author Response

(The authors gave the same response as above.)

Round 2

Reviewer 1 Report

The author addressed the comments fine and the manuscript can now be considered for publication.

Author Response

Thanks for the comments.